# Meta-Album: Multi-domain Meta-Dataset for Few-Shot Image Classification

**Ihsan Ullah**[*], **Dustin Carrión-Ojeda**[*¶‖], **Sergio Escalera**[#%], **Isabelle Guyon**[*%], **Mike Huisman**[+],
**Felix Mohr**[‡], **Jan N. van Rijn**[+], **Haozhe Sun**[*], **Joaquin Vanschoren**[§], **Phan Anh Vu**[*]

% ChaLearn, USA
‖ hessian.AI, Germany
# Universitat de Barcelona, Spain
‡ Universidad de La Sabana, Colombia
¶ Technische Universität Darmstadt, Germany
∗ LISN/CNRS/INRIA, Université Paris-Saclay, France
§ TU/e Eindhoven University of Technology, The Netherlands
+ Leiden Institute of Advanced Computer Science (LIACS), Leiden University, the Netherlands
https://meta-album.github.io/

## Abstract

We introduce Meta-Album, an image classification meta-dataset designed to facilitate few-shot learning, transfer learning, meta-learning, among other tasks. It includes 40 open datasets, each having at least 20 classes with 40 examples per class, with verified licences. They stem from diverse domains, such as ecology (fauna and flora), manufacturing (textures, vehicles), human actions, and optical character recognition, featuring various image scales (microscopic, human scales, remote sensing). All datasets are preprocessed, annotated, and formatted uniformly, and come in 3 versions (Micro ⊂ Mini ⊂ Extended) to match users' computational resources. We showcase the utility of the first 30 datasets on few-shot learning problems. The other 10 will be released shortly after. Meta-Album is already more diverse and larger (in number of datasets) than similar efforts, and we are committed to keep enlarging it via a series of competitions. As competitions terminate, their test data are released, thus creating a rolling benchmark, available through OpenML.org. Our website https://meta-album.github.io/ contains the source code of challenge winning methods, baseline methods, data loaders, and instructions for contributing either new datasets or algorithms to our expandable meta-dataset.[1]

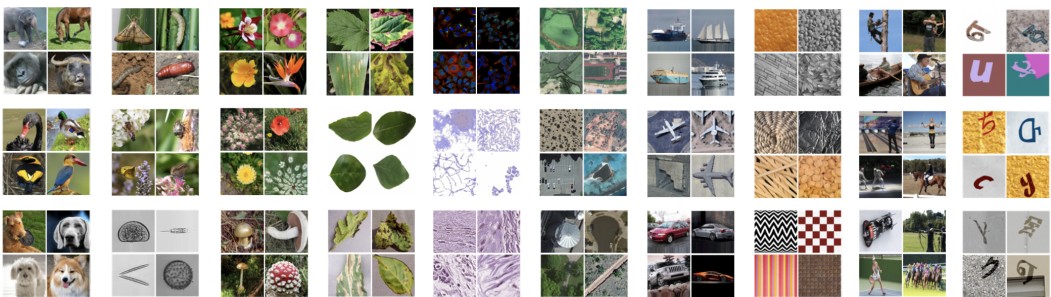

Figure 1: Meta-Album sample images. Each column represents one domain and each row one set. Domains are arranged in the same order as in Table 2.

---

[1]All authors except for the first two authors (equal contributions) are in alphabetical order of last name.

36th Conference on Neural Information Processing Systems (NeurIPS 2022) Track on Datasets and Benchmarks.

# 1 Introduction

## 1.1 Background

Machine learning has progressed rapidly in recent years and has enabled breakthroughs in various domains. The success of most machine learning techniques hinges on the availability of large amounts of data [34, 63], limiting their applicability in domains where only little data is available. Enabling machine learning algorithms to learn new tasks from only a few examples is studied within the field of *few-shot learning* [43, 57, 65]. Novel meta-learning algorithms have recently been proposed targeting few-shot learning, triggering a surge of popularity for such problems [4, 22, 26, 75]. Despite the popularity of the field, progress is held back by a lack of good, challenging, and computationally feasible meta-datasets, that enable us to accurately assess the generalization abilities of few-shot learning algorithms. To remedy this, we introduce **Meta-Album** (Figure 1), an extensible **multi-domain meta-dataset**, including (so far) 40 image classification datasets from 10 different domains: 30 of these are currently available through our website, and the remaining 10 will be released in spring 2023. This is part of a long-term effort to create a publicly available and growing meta-dataset, in conjunction with a meta-learning challenge series (the 2021 and 2022 editions are both parts of the NeurIPS competition track [6, 12, 13]). As competitions terminate, older datasets get released, thus refreshing a rolling benchmark, made available through OpenML.org [71]. We check that all datasets are free for use in academic research and provide their original licenses.

Meta-Album was specifically designed to facilitate meta-learning research in the cross-domain few-shot setting, which is more realistic than commonly used evaluation protocols. Traditionally, few-shot learning algorithms (*e.g.,* [14, 24, 61]) have been evaluated by taking an existing benchmark dataset from a particular "domain" (*e.g.,* handwriting recognition) with a large number of classes, and then breaking it down into smaller classification **tasks**, each including a *random* subset of classes (*e.g.,* a few specific characters). Algorithms are then tested for their ability to solve such tasks "quickly" from a small number of examples, after being trained on many other tasks. Typically the number of classes $N$ and examples per class $k$ are both *fixed* in what is known as an *N-way k-shot* learning problem. While this setting has served research well, it is not very representative of practical real-world applications where tasks may come from various domains, include classes not drawn at random, but stemming from a class hierarchy, and include any number of classes and/or examples per class. By providing data from a wide variety of domains, including datasets with many classes and a minimum number of examples per class, and retaining class hierarchy annotations, Meta-Album enables benchmarking according to a variety of more realistic settings.

## 1.2 Related work

In this section, we review meta-datasets previously proposed to benchmark few-shot learning and meta-learning, as well as large-scale multi-class datasets, and then contrast them with Meta-Album.

**Single dataset benchmarks**: Omniglot [32] is often used as a starting benchmark for few-shot learning and meta-learning. MiniImageNet [73] and Tiered-ImageNet [54] are adapted for few-shot image classification from ImageNet [56]. CIFAR-FS [2] and FC100 [45] are remodeled from CIFAR-100 [30] for few-shot settings.

**Multi-dataset benchmarks**: A recent trend is to assemble numerous datasets from different domains in the same benchmark. Visual Decathlon [53] gathers 10 diverse datasets. The focus is on finding a model with a universal representation capacity for use in many tasks. VTAB (Visual Task Adaptation Benchmark) [82] assembles 19 image classification tasks across various domains. These tasks are grouped into 3 partitions: natural, specialized, and structured. Meta-Dataset [70] includes 10 image classification datasets from several application domains in one collection. Meta-Dataset also leverages the label hierarchy in ImageNet and Omniglot to organize the tasks.

**Transfer learning and meta-learning benchmarks**: VTAB + MD [10] attempts to unify common transfer and meta-learning datasets in a single benchmark. The authors also provide a comparison of popular meta- and transfer learning methods. BSCD-FSL (Broader Study of Cross-Domain Few-Shot Learning) [20] gathers 4 real-world tasks to compare few-shot, meta-learning, and transfer learning methods. WILDS [28] is a benchmark of 10 datasets of various modalities (images, graphs, and text; 6 of them are image datasets), reflecting a diverse range of distribution shifts that naturally arise in real-world applications, and hence useful to evaluate meta-learning and transfer learning techniques.

Table 1: Comparison between Meta-Album and other large-scale or (meta-) datasets

| Dataset/ Meta-Dataset | # of domains | # of datasets | # of images | min/max classes per domain | min/max images per class | size on disk | multi-domain | lightweight (<20GB) | uniform # of images per class | uniform image size | repeated extensions |
|---|---|---|---|---|---|---|---|---|---|---|---|
| Meta-Dataset | 7 | 10 | 53 068 000 | 43/1 696 | 3/140 000 | 210 GB | ✓ | ✗ | ✗ | ✗ | ✗ |
| VTAB | 3 | 19 | 2 244 000 | 2/397 | 40/1 000 | 100 GB | ✓ | ✗ | ✗ | ✗ | ✗ |
| MS-COCO | 1 | 1 | 328 000 | 80/80 | 9/10 777 | 44 GB | ✗ | ✗ | ✗ | ✗ | ✗ |
| Mini Imagenet | 1 | 1 | 60 000 | 100/100 | 600/600 | 1 GB | ✗ | ✓ | ✓ | ✓ | ✗ |
| Omniglot | 1 | 1 | 32 000 | 1 623/1 623 | 20/20 | 148 MB | ✗ | ✓ | ✓ | ✓ | ✗ |
| CUB-200 | 1 | 1 | 6 000 | 200/200 | 20/39 | 647 MB | ✗ | ✓ | ✗ | ✗ | ✗ |
| CIFAR-100 | 3 | 1 | 60 000 | 15/50 | 600/600 | 161 MB | ✗ | ✓ | ✓ | ✓ | ✗ |
| **Meta-Album** *Micro* | **10** | **40** | **32 000** | **19/20** | **40/40** | **380 MB** | ✓ | ✓ | ✓ | ✓ | ✓ |
| **Meta-Album** *Mini* | **10** | **40** | **220 950** | **19/706** | **40/40** | **3.9 GB** | ✓ | ✓ | ✓ | ✓ | ✓ |
| **Meta-Album** *Extended* | **10** | **40** | **1 583 624** | **19/706** | **1/187 384** | **15 GB** | ✓ | ✓ | ✗ | ✓ | ✓ |

CTrL is a continual transfer-learning benchmark [72] including 7 commonly used datasets in image classification. In reinforcement learning, sets of simulation environments exist for meta- and transfer learning, such as Meta-World [81].

**Outside few-shot and meta-learning**: The AutoDL challenge [40] features a series of 66 datasets from numerous domains. These datasets cover a wide range of modalities: image, video, audio, text, and tabular. This competition focuses on finding a universal algorithm, which can solve many tasks without human supervision.

We compare Meta-Album with previous benchmarks/datasets in Table 1, and provide further details in Appendix G. Meta-Album covers a variety of domains, including ecology, manufacturing, textures, object classification, and character recognition, as well as a variety of scales: microscopic, macroscopic (human scale), or distant (remote sensing). While mostly re-purposing public datasets from heterogeneous sources to maximally vary recording conditions, we also introduce a few fresh datasets in OCR and ecology domains. Meta-Album comprises 3 different versions, Micro ⊂ Mini ⊂ Extended: **Micro** includes 20 classes and 40 images per class for ease of running sample code, **Mini** retains all original classes but also includes only 40 examples per class, while **Extended** includes all classes and examples. The variety of versions positions Meta-Album anywhere amongst small-scale datasets such as Omniglot [32], miniImageNet [73, 52] and CUB [74], which usually have at most 70 000 images in total and weigh at most a few GB, or very large-scale benchmarks such as Meta-dataset [70] and VTAB [10], which have more than 50 million images, weigh at least a few hundreds GB, and require high-end super-computer clusters. Its principal distinguishing feature is that it has, by far, the **largest number of domains and datasets**, collected in different conditions, and that it is designed to be **continually extended by either adding new domains or new datasets** in already existing domains, making it a tool of choice for cross-domain, domain-independent, and continual learning studies. Secondly, while other benchmarks usually provide only raw data, we **format all images uniformly as** $128 \times 128$ **pixel maps**, which has two benefits: reducing the storage/memory footprint and facilitating the benchmarking of methods independent of preprocessing steps. To that end, we optimized cropping and resizing to reduce dimensions as much as possible without degrading performance too much. In addition, Meta-Album includes datasets that have a **large number of classes** and class hierarchy annotations when available, with a **minimum number of classes and examples per class**: at least 20 classes (except one dataset having only 19 classes) with a minimum of 40 examples per class. This facilitates benchmark design, allowing us to vary the number of classes and the number of training examples per class over a large range of values. Finally and importantly, we selected datasets that are **not typically used in transfer-learning or meta-learning benchmarks**, *e.g.,* for pre-training backbone networks, such as ImageNet (which is included in *e.g.,* Meta-Dataset), or for conducting other meta-learning or transfer-learning experiments, such as Omniglot, CIFAR-100, SVHN, or MNIST (which are included in *e.g.,* VTAB and CTrL). This avoids giving an unfair advantage to methods that were developed using such commonly used datasets.

### 1.3 Contributions and recommended use

In summary, the contributions of our work are the following.

- We provide a **new meta-dataset for few-shot learning and meta-learning** consisting of 40 uniformly formatted datasets from 10 domains, which facilitates research in cross-domain meta-learning as well as practical and realistic evaluation of few-shot algorithms.

- We provide 3 versions of each dataset: Micro, Mini, and Extended to **facilitate usage by researchers with access to different amounts of computational power**.

- We **uniformly preprocessed and formatted data**, but also provide **instructions to retrieve the corresponding raw data** on our aforementioned website.

- We stimulate **community-driven benchmarking**, in conjunction with our challenge series, by welcoming new contributors and providing software and instructions to create additional datasets for Meta-Album, with strict quality control and review processes.

- We showcase our new meta-dataset by performing an **experimental evaluation** for several use cases, including transfer learning, few-shot meta-learning, and cross-domain few-shot meta-learning tasks, using a variety of algorithms, and we **open-source the code used**.

The recommended use of Meta-Album is to conduct fundamental research on machine learning algorithms and perform benchmarks, particularly in few-shot learning, meta-learning, continual learning, transfer learning, and image classification. Meta-Album is not recommended to create products, whether commercial or not, or to derive scientific findings outside benchmarking.

## 2 Meta-Album design and initial release

In this section, we explain the motivations behind the design of Meta-Album and present the 30 datasets included in the initial NeurIPS 2022 release. 10 more datasets are kept private, and will be released in spring 2023.

### 2.1 Motivation

Meta-Album emerged from a sequence of few-shot meta-learning benchmarks, following the problem formulations described in Section 3.1. The first of these was the 2020 MetaDL-mini challenge, which was run in conjunction with AAAI 2021 [12]. It followed the "within domain few-shot learning" protocol, and algorithms were evaluated with small-scale public datasets (Omniglot and CIFAR-100). Subsequently, we designed a first version of Meta-Album, including 15 datasets, for a larger-scale "within domain few-shot learning" challenge (MetaDL @ NeurIPS 2021 [13]). Here, algorithms were meta-trained and meta-tested on tasks extracted from a single dataset at a time, and performances were averaged over 5 datasets, both in the feedback phase and the final evaluation phase, to obtain a more robust evaluation. The 5 extra datasets were provided for practice purposes. The results of this challenge (further detailed in Section 3) indicated that these tasks were well within reach of state-of-the-art methods. This motivated us to move to the "cross-domain few-shot learning" setting. The design of this new challenge (part of an official NeurIPS 2022 challenge [6]) motivated us to grow Meta-Album to 30 datasets spanning multiple domains. We intend to continue growing Meta-Album and already have 10 more datasets lined up, in preparation for the next challenge. This will constitute a *rolling benchmark*: with each new challenge, previous feedback datasets are publicly released, previous final evaluation datasets become feedback datasets, and fresh datasets become final evaluation datasets.

Existing meta-datasets did not allow us to carry out our challenge program for several reasons: (1) they included datasets too familiar to the meta-learning community; (2) they did not include enough datasets to robustly evaluate participants (particularly in the cross-domain setting); (3) their datasets had a large variance in number of classes and examples per class, introducing bias in our experimental design. This required us to source new datasets. Furthermore, since these challenges include code submission, and providing the same resources to all participants, we needed to limit computational resources. Therefore, we had to downscale images while taking care that this does not significantly degrade performance.

## 2.2  Data search

Many people were involved in the sourcing of all datasets and their preparation, and they are gratefully acknowledged in our acknowledgements. This collaboration followed precise instructions to identify datasets that are: (i) from the same domain; (ii) freely available for academic research; (iii) having at least 20 classes with at least 40 examples per class; (iv) with images of good enough quality by visual inspection and with no offensive material (we excluded "deprecated" datasets); (v) with baseline performance within a given range.

The last criterion was needed to ensure the success of our challenges, since tasks that are too easy or too hard do not allow us to separate challenge participants. For the purpose of designing Meta-Album, we defined a "domain" according to four characteristics: (1) application domain; (2) pattern recognition problem (texture or object classification); (3) scale: micro, human scale, or distant; (4) input channels. We ended up with 10 domains (see Table 2): Large animals, small animals, plants, plant diseases, microscopy, remote sensing, vehicles, manufacturing, human actions, and optical character recognition (OCR). Data sources were very varied, and mostly came from internet searches, but we also produced our own optical character recognition datasets and obtained novel donated data.

## 2.3  Data preparation

We performed several iterations of preprocessing, experiments, and analyses to prepare the datasets. This workflow included identifying and, when possible, correcting bias and artifacts (including artifacts we may have introduced by resizing and cropping images), and making sure that images are recognizable by human eye inspection.

As we work with datasets from diverse sources, each dataset requires a different preprocessing strategy, *e.g.,* the small animals' datasets, plant-diseases datasets, manufacturing, and remote sensing datasets have images in different resolutions and orientations. However, usually, the object of interest lies in the middle of the image, which facilitated cropping images horizontally or vertically to get squared images. In some cases, *e.g.,* the plankton dataset, the image orientation depends on the shape of the plankton and the way it is photographed, *i.e.,* images have vertical or horizontal orientations based on the plankton in the image. Cropping images is not useful in this case because a big part of the plankton would be cropped. As such, we added a matching background to the images (either horizontally or vertically) by extending the top and bottom 3 rows or left and right columns respectively. In order to make sure that we do not introduce artifacts in the data, afterwards we applied a Gaussian kernel of size (29, 29) using open-cv [3] to the newly constructed background. In other cases, the area of interest was not necessarily centered, *e.g.,* human action datasets, and we had to use a human face detector to locate the subject, and then we cropped the upper body. For all datasets except for the optical character recognition datasets, we resized the images to a $128 \times 128$ resolution using an anti-aliasing filter [3]. The optical character recognition datasets are synthetically generated directly to the correct dimension by OmniPrint [64] (MIT license), and do not need further processing. The preprocessed data was formatted in a data format conserving as much meta-data as possible. For the micro and the mini version, only classes with at least 40 examples are kept for each dataset to maintain a balance between a large number of classes and sufficient examples per class while for the extended version, all classes and all images are kept. More details about data preparation and formatting can be found in Appendix C.

## 2.4  Initial Meta-Album release

The initial release of Meta-Album consists of 3 datasets for each of the 10 domains. Each dataset has 3 versions controlling the size, as explained in Section 1.2. All datasets are annotated with class labels and other meta-data. All 30 datasets were chosen after careful and critical analysis during the data preparation and quality control steps as described in Appendix C. Table 2 provides statistics on the various versions; Figure 1 shows sample images from each dataset. More details about datasets and their meta-data are listed in Appendix A. License information for all datasets can be found in Appendix B. Meta-Album datasets are being used in the NeurIPS Cross-domain meta-learning Challenge 2022. The first 30 datasets are available on OpenML [71], and later in spring 2023, 10 more datasets will be released, followed by other releases as our challenge program unfolds. Details about how to access Meta-Album datasets, contribute to the open meta-dataset, prepare new datasets with quality control, and submit these datasets for inclusion in Meta-Album can be found on the

Table 2: Meta-Album: Datasets summary (*Mini versions*)

| Domain ID | Domain Name | Set # | Dataset ID | Dataset Name | # Categories | # Images | Original source |
|---|---|---|---|---|---|---|---|
| LR_AM | Large Animals | 0 | BRD | Birds | 315 | 12 600 | Birds 400 [50] |
| | | 1 | DOG | Dogs | 120 | 4 800 | Stanford Dogs [27] |
| | | 2 | AWA | Animals with Attributes | 50 | 2 000 | AWA [78] |
| SM_AM | Small Animals | 0 | PLK | Plankton | 86 | 3 440 | WHOI [62] |
| | | 1 | INS_2 | Insects 2 | 102 | 4 080 | Pest Insects [76] |
| | | 2 | INS | Insects | 104 | 4 160 | SPIPOLL [59] |
| PLT | Plants | 0 | FLW | Flowers | 102 | 4 080 | Flowers [44] |
| | | 1 | PLT_NET | PlantNet | 25 | 1 000 | PlantNet [18] |
| | | 2 | FNG | Fungi | 25 | 1 000 | Danish Fungi [48] |
| PLT_DIS | Plant Diseases | 0 | PLT_VIL | PlantVillage | 38 | 1 520 | PlantVillage [23, 46] |
| | | 1 | MED_LF | Medicinal Leaf | 25 | 1 000 | Medicial Leaf [55] |
| | | 2 | PLT_DOC | PlantDoc | 27 | 1 080 | Plant Doc [60] |
| MCR | Microscopy | 0 | BCT | Bacteria | 33 | 1 320 | DiBas [84] |
| | | 1 | PNU | PanNuke | 19 | 760 | PanNuke [16, 17] |
| | | 2 | PRT | Subcel. Human Protein | 21 | 840 | Protein Atlas [66] |
| REM_SEN | Remote Sensing | 0 | RESISC | RESISC | 45 | 1 800 | RESISC45 [8] |
| | | 1 | RSICB | RSICB | 45 | 1 800 | RSICB128 [35] |
| | | 2 | RSD | RSD | 38 | 1 520 | RSD46 [79, 41] |
| VCL | Vehicles | 0 | CRS | Cars | 196 | 7 840 | Cars [29] |
| | | 1 | APL | Airplanes | 21 | 840 | Multi-type Aircraft [77] |
| | | 2 | BTS | Boats | 26 | 1 040 | MARVEL [19] |
| MNF | Manufacturing | 0 | TEX | Textures | 64 | 2 560 | KTH-TIPS [15, 42] Kylberg [31] UIUC [33] |
| | | 1 | TEX_DTD | Textures DTD | 47 | 1 880 | Texture DTD [9] |
| | | 2 | TEX_ALOT | Textures ALOT | 250 | 10 000 | Texture ALOT [5] |
| HUM_ACT | Human Actions | 0 | SPT | 100 Sports | 73 | 2 920 | 100 Sports [49] |
| | | 1 | ACT_40 | Stanford 40 Actions | 39 | 1 560 | Stanford 40 Actions [80] |
| | | 2 | ACT_410 | MPII Human Pose | 29 | 1 160 | MPII Human Pose [1] |
| OCR | Optical Char. Recog. | 0 | MD_MIX | OmniPrint-MD-mix | 706 | 28 240 | |
| | | 1 | MD_5_BIS | OmniPrint-MD-5-bis | 706 | 28 240 | OmniPrint [64] |
| | | 2 | MD_6 | OmniPrint-MD-6 | 703 | 28 120 | |

Meta-Album Website. This web page will also inform on software updates and revisions or new releases of our meta-dataset.

# 3 Use cases and baselines

This section illustrates how Meta-Album can be used for a variety of purposes. The code of all experiments is provided in our GitHub repository https://github.com/ihsaan-ullah/meta-album, and can serve as a basis to benchmark new algorithms against the baseline methods we investigate here. The problems investigated range from few-shot learning (for which Meta-Album was designed) to multi-class image classification, transfer learning, hierarchical classification, and continual learning. Because of lack of space, we only report few-shot learning experiments.

## 3.1 Problem setting

In this paper, we focus on **few-shot image classification**, where the goal is to **learn to perform new classification tasks from a limited number of examples**. Here, every task $\mathcal{T}_j = (\mathcal{D}_{\mathcal{T}_j}^{train}, \mathcal{D}_{\mathcal{T}_j}^{test})$ consists of a *support set* $\mathcal{D}_{\mathcal{T}_j}^{train}$ with training examples and a *query set* $\mathcal{D}_{\mathcal{T}_j}^{test}$ with test examples.[2] In $N$-way $k$-shot classification, we require that every *support set* contain exactly $N$ classes with $k$ examples per class ($kN = |\mathcal{D}_{\mathcal{T}_j}^{train}|$). Another requirement is that the classes in the query set must occur in the support set.

Few-shot learning does not necessarily require meta-learning. As in other "regular" learning problems, a *learner*, having available a set of training examples $\mathcal{D}_{\mathcal{T}_j}^{train}$ for a given task, can just return a *trained model* (classifier). But meta-learning is frequently used to enhance few-shot learning.

In a meta-learning problem, a *meta-learner*, having available a set of $m$ training *tasks* $\mathcal{M}_{\mathcal{D}}^{train} = \{\mathcal{T}_j\}_{j=1}^m$, returns a meta-trained *learner*. In order to develop a meta-trained few-shot *learner*, available data organized in tasks $\mathcal{M}_{\mathcal{D}}$ (coming either from one or multiple datasets) are split into three "meta-splits" containing *disjoint sets of classes*: *meta-training* split $\mathcal{M}_{\mathcal{D}}^{train}$, *meta-validation* split $\mathcal{M}_{\mathcal{D}}^{valid}$,

---

[2]The nomenclature *support set* instead of *training set*, and *query set* instead of *test set* is common in the meta-learning literature. It highlights the fact that, while meta-training is done on *tasks = {support set, query set}*, no actual test-data is presented to the classifier. The meta-test data also includes pairs of support and query sets, from which the ground truth of query set samples is hidden from the classifier.

Table 3: Datasets used in the NeurIPS 2021 MetaDL challenge [13].

| Phase | Datasets according to Table 2 |
|---|---|
| Feedback Phase | SM_AM.PLK, MDN.MLD, MNF.TEX_DTD, REM_SEN.RSICB, OCR.MD_MIX |
| Final test phase | SM_AM.INS, PLT_DIS.PLT_VIL, MNF.TEX, REM_SEN.RESISC, OCR.MD_5_BIS |

and *meta-testing* split $\mathcal{M}_{\mathcal{D}}^{test}$. The *learner* is meta-trained with $\mathcal{M}_{\mathcal{D}}^{train}$. During meta-training, the *learner* is evaluated with $\mathcal{M}_{\mathcal{D}}^{valid}$ every few meta-training cycles, to monitor progress. The final product of meta-training when the time budget has elapsed, is the *learner* with the highest performance on $\mathcal{M}_{\mathcal{D}}^{valid}$ tasks. It is then evaluated on tasks from $\mathcal{M}_{\mathcal{D}}^{test}$.

Within the realm of few-shot learning, we distinguish two cases. **Within domain few-shot learning** refers to the problem where data from the meta-validation and meta-test splits come from the same domain as meta-training data. Here, domain refers to one single dataset of Meta-Album $\mathcal{D}_i$, $i \in \{1, \ldots, 30\}$. We enforce that $\mathcal{D}_i$ is partitioned into $\mathcal{M}_{\mathcal{D}_i}^{train}$, $\mathcal{M}_{\mathcal{D}_i}^{valid}$, and $\mathcal{M}_{\mathcal{D}_i}^{test}$, using three disjoint sets of classes. In this setting, the goal of *learners* is to learn tasks including classes coming from the same original domain/dataset. If the *learner* has been meta-trained, **test tasks include new classes unseen during meta-training**. **Cross-domain few-shot learning**, in contrast, is a setting for which meta-split is performed at *dataset level* instead of *class level*. Once the *learner* has been meta-trained, **test tasks come from new datasets unseen during meta-training**. Note that as a consequence, there is still a slight domain overlap between the meta-train, meta-validation and meta-test dataset. For example, the meta-train dataset can contain observations from ten different datasets, including the 'Fungi' dataset, whereas the learner will be evaluated on a meta-test dataset constructed from ten different datasets, including the 'Flowers' dataset. This introduces two important challenges for the meta-learning algorithms whenever confronted with a given task in the meta-test set: 1) it has to deal with several classes in the meta-train set that are not related to the concepts from the task at hand. 2) while there are indeed observations in the meta-train set that are related to concept of the current task, these come from a different dataset, and might be sampled according to different conditions (different camera, lightning, geographical area, etc.). This aligns with the cross-domain setting introduced in the NeurIPS'22 meta-learning challenge [6]. Beyond the cross-domain setting, one can imagine a 'domain independent' setting, where each of the meta-train, meta-validation and meta-test datasets contain classes from different domains, and therefore no domain knowledge from the meta-train phase can be exploited.

We also distinguish between **fixed N-way k-shot** evaluations and **any-way any-shot** evaluations. The former requires fixing the value of $N$ and $k$ for the entire benchmark. The latter requires randomly choosing $N$ and $k$ for each task, within pre-defined ranges. Meta-Album allows us to choose $N \in [2, 20]$ and $k \in [1, 20]$.

## 3.2 Experiments

The first motivational use of Meta-Album has been the NeurIPS 2021 MetaDL challenge [13]. This was a meta-learning challenge with code submission, aiming at evaluating **few-shot learning methods in the within domain setting**, as described in Section 3.1. The evaluation was carried out with 600 tasks in the **5-way 5-shot setting**, using a subset of Meta-Album (see Table 3).

The solutions of the top participants have been open-sourced. In a paper, authored collaboratively between the competition organizers and the top-ranked participants [13], we analyse the results of the competition. The lessons learned include that learning good representations is essential for effective transfer learning. The winner's solution MetaDelta++ [7], based on a combination of pre-trained backbone networks, performed best on all final 5 test phase datasets, with high accuracy scores (0.98, 0.94, 0.99, 0.92, 0.94). This indicates that, in future challenges, we are ready to tackle harder tasks, and motivated us to move to **cross-domain few-shot learning**, in the **any-way any-shot setting** for the NeurIPS 2022 challenge [6]. Fine-tuning backbones on meta-training data turned out to be important, though there are indications that off-the-shelf backbones pre-trained with self-supervised learning on massive datasets might become the way of the future, essentially making meta-learning unnecessary for image classification problems. Thus, meta-learning should be benchmarked in **de**

**novo training conditions**, in the future, to prepare for scenarios (in other domains) in which such backbones are not available. The NeurIPS 2022 challenge encourages *de novo* training in a dedicated league. Appendix D contains a detailed analysis of the difficulty of all Meta-Album datasets following the NeurIPS 2021 MetaDL challenge protocol.

**Difficulty of cross-domain few-shot learning**

To evaluate the gap in difficulty between "within domain" and "cross-domain" few-shot learning problems (Section 3.1), we carried out first experiments in the 5-way [1, 5, 10, 20]-shot setting. For all experiments, we use Meta-Album Mini, single PNY GeForce RTX 2080TI GPUs with 11GB of VRAM or a single NVIDIA V100 with 16GB of VRAM. Each experimental run took at most 24 hours on the former GPU (for details, please see Appendix E and Appendix F).

Although several methods have been proposed in the state-of-the-art to tackle the cross-domain few-shot learning problem [11, 36, 37, 38, 39, 51, 69], they require too much time or are not compatible with our fully supervised setting. Therefore, we investigated the few-shot learning performance of popular meta-learning methods: MAML [14], Matching networks [73], and Prototpical networks [61]. We compared them against two baseline methods: TrainFromScratch (learning every task starting from a random initialization at meta-test time, *i.e.,* no meta-learning) and FineTuning, which is pre-trained on the classification problem arising from concatenating all meta-training classes and corresponding data and only fine-tunes the last layer at meta-test time [7]. All techniques use a ResNet-18 backbone [21] and are trained from scratch on Meta-Album (not using any pre-trained feature extractors) using the best-reported hyperparameters by the original authors on 5-way 5-shot miniImageNet (*i.e.,* for FineTuning the backbone is pre-trained with Meta-Album meta-training data only). It is worth mentioning that the purpose of our baseline methods is to give a set of "classical" and "representative" techniques, not to be exhaustive.

For a given dataset, all meta-learning techniques are meta-trained on $60\,000$ tasks. However, the backbone used for FineTuning is meta-trained (pre-trained) on $60\,000$ randomly sampled batches of size 16. The performance of trainers is validated every $2\,500$ tasks (or batches in case of pre-training the FineTuning backbone). The query set for every task contains 16 examples per class, following [7]. The learning algorithm with the best validation performance is evaluated on 600 meta-test tasks randomly sampled from the meta-testing split, which has information from unseen classes during training and validation. We average the results over 3 runs with different random seeds. Error bars are $95\%$ confidence intervals of the mean overall meta-test tasks in all runs ($1\,800$ tasks per dataset).

Results are shown in Figure 2. A first observation is that Prototypical Networks (ProtoNet) dominate other algorithms (both within domain and cross-domain) and that the ranking of algorithms does not significantly change with the number of shots. However, the exception is FineTuning for 1-shot learning in the cross-domain configuration, which outperforms ProtoNet by a small margin. Moreover, we observe that FineTuning outperforms MAML and Matching Networks (the other episodic meta-learning algorithm we tried), corroborating findings showing that finetuning yields excellent few-shot learning performance without using episodic meta-learning [7, 25, 67, 68]. We also see that the naive baseline TrainFromScratch yields the worst performance, indicating that meta-learning actually helps transfer knowledge to new tasks. Furthermore, we observe that the performances improve with the number of shots (training examples per class). Lastly, the details provided in Appendix E and Appendix F show that FineTuning is the fastest method at training time while ProtoNet and MatchingNet are the fastest methods at inference time with less than 1 second per task.

For cross-domain few-shot learning, as can be expected, the accuracy is lower since the problem is more complex. However, it does not dramatically decrease compared to within domain few-shot learning, which leads us to speculate that such a new problem is within reach of the current state-of-the-art. This gives rise to new opportunities for improvement in this more complicated and more realistic setting.

**Difficulty of "any"-way "any"-shot learning**

Moving to yet more realistic and harder tasks, we also investigated the performance in the "any"-way "any"-shot setting, where tasks at meta-test time include a varying number of classes between 2 to 20 and a varying number of examples per class between 1 to 20. For example, at meta-test and meta-validation time, some carved out tasks might be as follows: **Test task 1:** 5-way 1-shot task

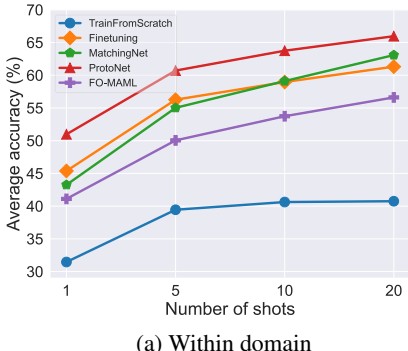

(a) Within domain

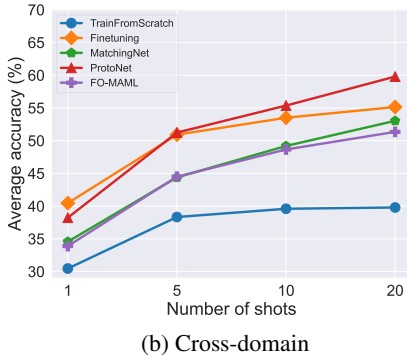

(b) Cross-domain

Figure 2: **Comparison of "within domain" and "cross-domain" few-shot learning.** We plot 5-way [1, 5, 10, 20]-shot learning meta-test mean task accuracy, averaged over 1 800 tasks drawn from the 30 released Meta-Album datasets. Corresponding 95% confidence intervals are within the size of the symbols.

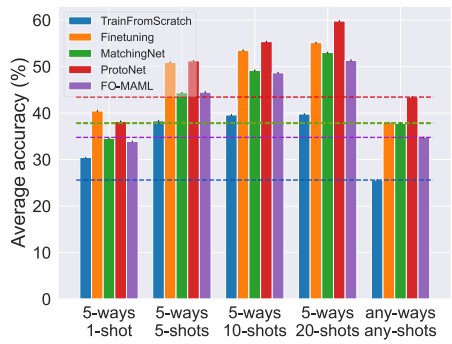

(a) Difficulty of fixed and variable # of ways and shots

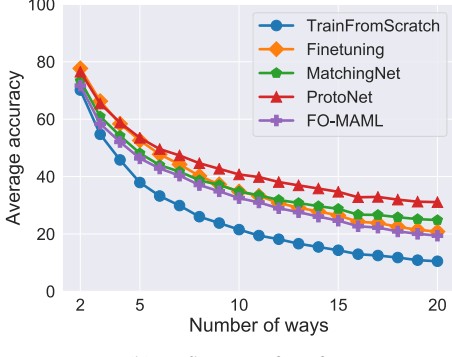

(b) Influence of # of ways

Figure 3: **Comparison of "cross-domain" few-shot learning using fixed and variable number of ways and shots, and influence of number of ways on performance.** We plot few-shot learning meta-test mean task accuracy, averaged over test tasks drawn from the 30 released Meta-Album datasets. Corresponding 95% confidence intervals are almost imperceptible as they are around $\pm 0.15$.

from Dataset 9; **Test task 2:** 3-way 15-shot task from Dataset 3; **Test task 3:** 12-way 4-shot task from Dataset 8; etc. However, during the meta-training phase, we kept the number of classes constant (specifically, we used 5-way any-shot tasks). This facilitates using off-the-shelf meta-learning techniques. All other experimental conditions (hyper-parameters, computational resources) are the same as in the previous section.

In Figure 3a we can observe that the complexity of the any-way any-shot setting is similar to the 5-way 1-shot setting. Nevertheless, the meta-learning approaches (ProtoNet, MatchingNet, MAML) adapt better to this novel setting since their performance is better than the one achieved in the 5-way 1-shot setting, while the performance of FineTuning and TrainFromScracth is worse compared to the same setting. Additionally, the results presented in Figure 3b and Appendix F show that the dominant difficulty factor in any-way any-shot learning is the variability in the number of ways since as it can be seen, the performance of the evaluated methods is highly affected by the increment in this number. This is supported by the fact that the absolute Pearson correlation between the number of ways and the test accuracy is larger (r=-0.55, p<0.05) than the correlation between the number of shots and the accuracy (r=0.1, p<0.05). Therefore, we anticipate that this new setting of any-way any-shot learning will deliver new interesting results in the upcoming challenge.

# 4   Discussion and conclusion

We introduce Meta-Album, a new meta-dataset for few-shot image classification, which is both practical and extensive: it includes many datasets from a wide variety of domains, all preprocessed to allow training according to different settings on commodity GPUs. It is especially amenable to evaluating meta-learning and transfer learning techniques. It can also be used for hierarchical classification as well as domain adaptation, due to the presence of overlapping classes between datasets, and continual learning, where algorithms are progressively trained across datasets.

We evaluate the utility of Meta-Album using a range of few-shot learning experiments. Our findings include that Prototypical Networks and the FineTuning baseline perform quite well. This corroborates the results of the NeurIPS'21 challenge, in which the winners capitalized on the use of pre-trained backbones, obtaining results in the high 90% classification accuracy in the "within domain" 5-way 5-shot setting [13]. Meta-Album will further challenge the research community by being considerably larger and by mixing tasks from multiple domains, in [2-20]-way [1-20]-shot settings. Furthermore, Meta-Album allows *de novo* training. We tested and compared this new framework to that of previous challenges and demonstrated an increased difficulty on all our baseline methods.

In preparing the datasets we identified several types of biases, including correlations between class labels and nuisance variables (*e.g.,* background, luminosity, contrast, colour spectrum, position and orientation of objects). In this first release, we avoided correcting such biases, to avoid introducing yet more bias, and opted to homogenize the datasets by shuffling the examples. We documented our findings to facilitate the creation of challenges that study the problem of bias, in which the (meta-)training data and (meta-)test data will have distribution shifts.

In future work, we want to make tasks more challenging by ensuring that every task consists of related concepts, belonging to a same super-class. For example, insects from the Coleoptera order have more resemblance with one another than with insects coming from another order, *e.g.,* butterflies. As such, it would be more challenging to have a task where the goal is to classify various insects from the Coleoptera order, rather than a tasks where insects from various orders are combined into one classification task. This requires datasets in which class hierarchies are provided, and we currently have only a few of those. Further work also includes introducing the even more difficult "domain independent" settings, in which meta-training and meta-testing are performed on entirely different domains. Indeed, a problem setting where the goal is to learn how to learn on tasks that are not related to the meta-test data, would truly challenge a meta-learning system.

## Acknowledgments and Disclosure of Funding

We gratefully acknowledge the data owners/creators:

**LR_AM.BRD**: Gerald Piosenka, *Scottsdale, Arizona, United States*; **LR_AM.DOG**: Aditya Khosla, Nityananda Jayadevaprakash, Bangpeng Yao and Li Fei-Fei from *Stanford University*; **LR_AM.AWA**: Christoph H. Lampert, Bernt Schiele and Zeynep Akata.

**SM_AM.PLK**: Heidi M. Sosik, Emily E. Peacock, Emily F. Brownlee and Eric Orenstein from *Woods Hole Oceanographic Institution, United States*; **SM_AM.INS**: Grégoire Loïs, Colin Fontaine and Jean-Francois Julien from *National Museum of Natural History Paris, France* and *SPIPOLL Science project*; **SM_AM.INS_2**: Xiaoping Wu, Chi Zhan, Yukun Lai, Ming-Ming Cheng and Jufeng Yang.

**PLT.FLW**: Maria-Elena Nilsback and Andrew Zisserman from *University of Oxford, England*; **PLT.PLT_NET**: Garcin Camille, Joly Alexis, Bonnet Pierre, Lombardo Jean-Christophe, Affouard Antoine, Chouet Mathias, Servajean Maximilien, Salmon Joseph and Lorieul Titouan; **PLT.FNG**: Lukáš Picek, Milan Šulc, Jiří Matas, Jacob Heilmann-Clausen, Thomas S. Jeppesen, Thomas Læssøe and Tobias Frøslev.

**PLT_DIS.PLT_VIL**: Sharada Mohanty, David Hughes, and Marcel Salathé, from *EPFL Switzerland* and *Penn State University*, J. Arun Pandian and G. Geetharamani, from *Department of Mathematics, University College of Engineering, Anna University - BIT Campus and Department of Computer Science and Engineering, M.A.M. College of Engineering and Technology, Tiruchirappalli, India*; **PLT_DIS.MED_LF**: S Roopashree, J Anitha, from *Visvesvaraya Technological University, R V Insti-*

*tute of Management, Dayananda Sagar University, India*; **PLT_DIS.PLT_DOC**: Sharada Mohanty, David Hughes, and Marcel Salathé.

**MCR.BCT**: Bartosz Zieliński, Anna Plichta, Krzysztof Misztal, Przemysław Spurek, Monika Brzychczy-Włoch and Dorota Ochońska from *Uniwersytet Jagielloński*; **MCR.PRT**: Peter J Thul, Lovisa Akesson, Mikaela Wiking, Diana Mahdessian, Aikaterini Geladaki, Hammou Ait Blal, Tove Alm, Anna Asplund, Lars Björk, Lisa Breckels, and others from *Protein Atlas*; **MCR.PNU**: Gamper Jevgenij , Koohbanani Navid Alemi , Benet Ksenija , Khuram Ali and Rajpoot Nasir from *University of Warwick*.

**REM_SEN.RESISC**: Gong Cheng, Junwei Han, and Xiaoqiang Lu from *Northwestern Polytechnical University, Xi'an, China*; **REM_SEN.RSICB**: Haifeng Li, Xin Dou, Chao Tao, Zhixiang Hou, Jie Chen, Jian Peng, Min Deng, Ling Zhao from *Central South University, Changsha, China*; **REM_SEN.RSD**: Yang Long, Yiping Gong, Zhifeng Xiao, and Qing Liu, Deren Li, Chunshan Wei, Gefu Tang and Junyi Liu from *State Key Laboratory of Information Engineering in Surveying Mapping and Remote Sensing, Wuhan University, Wuhan 430079, China*.

**VCL.CRS**: Jonathan Krause, Michael Stark, Jia Deng and Li Fei-Fei from *Stanford University*; **VCL.APL**: Wu Zhize; **VCL.BTS**: Gundogdu E., Solmaz B, Yucesoy V., Koc A.

**MNF.TEX**: Eric Hayman, Barbara Caputo, Mario Fritz, P. Mallikarjuna and Alireza Tavakoli Targhi from *KTH Royal Institute of Technology in Stockholm* (for KTH TIPS and KTH TIPS 2); Gustaf Kylberg from *Uppsala University, Sweden* (for Kylberg Texture); Jean Ponce, Svetlana Lazebnik and Cordelia Schmid from *University of Illinois Urbana-Champaign* (for UIUC Textures); **MNF.TEX_DTD**: Mircea Cimpoi, Subhransu Maji, Iasonas Kokkinos, Sammy Mohamed and Andrea Vedaldi, the Authors of Describable Textures Dataset (DTD); **MNF.TEX_ALOT**: Gertjan Burghouts and Jan-Mark Geusebroek from *University of Amsterdam, Netherlands*.

**HUM_ACT.SPT**: Gerald Piosenka, *Scottsdale, Arizona, United States*; **HUM_ACT.ACT_40**: B. Yao, X. Jiang, A. Khosla, A.L. Lin, L.J. Guibas, and L. Fei-Fei from *Stanford University*; **HUM_ACT.ACT_410**: Mykhaylo Andriluka and Leonid Pishchulin and Peter Gehler and Schiele Bernt.

**OCR.MD_MIX, OCR.MD_5_BIS, OCR.MD_6**: Generated by Haozhe Sun (co-author).

We acknowledge the efforts of Philip Boser, Maria Belen Guaranda Cabezas, Jilin He, Felix Heron, Gabriel Lauzzana, Romain Mussard, and Manh Hung Nguyen for datasets and datasheets preparation. We also received useful input from many members of the TAU team of the LISN laboratory, Wei Wei Tu from 4Paradigm Inc, China, and the MetaDL technical crew: Adrian El Baz, Zhengying Liu, Adrien Pavao, Jennifer (Yuxuan) He, Yui Man Lui, Sébastien Treguer, Benjia Zhou, and Jun Wan, who participated in identifying datasets and contributed to discussions. We also would like to thank the hundreds of volunteers involved in the SPIPOLL citizen science program who pictured and identified insects.

This work was supported by ChaLearn, the ANR (Agence Nationale de la Recherche, National Agency for Research) under AI chair of excellence HUMANIA, grant number ANR-19-CHIA-0022 and Labex Digicosme project ANR11LABEX0045DIGICOSME operated by ANR as part of the program Investissement d'Avenir Idex Paris Saclay (ANR11IDEX000302). In addition, some experiments were performed using the compute resources from the Academic Leiden Interdisciplinary Cluster Environment (ALICE) provided by Leiden University. This research was partially supported by TAILOR, a project funded by EU Horizon 2020 research and innovation programme under GA No 952215.

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
