# OpenReview forum: "Meta-Album: Multi-domain Meta-Dataset for Few-Shot Image Classification"
_NeurIPS.cc/2022/Track/Datasets_and_Benchmarks — NeurIPS 2022 Datasets and Benchmarks _

### Official Review · Reviewer_FXCy · 2022-07-10
**Comments**

**Rating:** 7
**Confidence:** 4
**Clarity:** The paper is very clear and well writ…

**Strengths:**

- The paper is well motivated that there’s lack of challenging but realistic cross-domain few-shot learning dataset and benchmark in the literature;
- The paper acquired a very large-scale and relatively-diverse few-shot learning dataset;
- Popular baseline methods and data loaders are well-prepared for users;
- The learning tasks (within-domain, cross-domain, fix-way fix-shot, and any-way any-shot) are well-defined with a range of different levels of difficulty;


**Weaknesses:**

- By providing multiple versions of the dataset, it might introduce difficulties for researchers to choose which one to use and to compare their study to existing works (which might leverage different versions);
- The domains in the existing dataset are relatively close to each other. The authors might want to introduce images from the medical domain such as histopathological images, x-ray, and MRI, to make the tasks even more challenging;


**Additional Feedback:**

N/A

**Correctness:**

The claims in the paper are valid and the dataset is constructed in a sound way. The experimental design and evaluation metrics are well defined.

**Documentation:**

The paper provides sufficient detail on data collection and organization, availability and maintenance, and ethical and responsible use.

**Ethics:**

There’s no ethical concerns for this work.

**Relation To Prior Work:**

The authors thoroughly discussed the relevant literature.

**Summary And Contributions:**

The authors provide a large-scale dataset and benchmark for multi-domain few-shot learning with 30 sub-datasets from 10 different domains. They have three versions (in scale) of data to meet the users’ needs. They also provide implemented baseline methods and data loaders for the challenge. In the paper, they also compared several baseline meta-learning methods such as MAML, matching networks, and prototypical networks in within-domain and cross-domain experimental settings for k-ways n-shot learning, and any-way any-shot learning tasks.

---

> ### Author Response · Authors · 2022-08-06
> **Reviewer's feedback addressed**
>
> We thank the reviewer for their valuable feedback.
>
> ### **Weaknesses:**
>
> **1. By providing multiple versions of the dataset, it might introduce difficulties for researchers to choose which one to use and to compare their study to existing works (which might leverage different versions);**
>
> We agree with this assessment. To mitigate your concern, we propose to put in place a versioning system for our datasets. We are currently studying various options and are open to suggestions made by the reviewers.
>
> We do see value in allowing multiple versions, though. First, large datasets like ImageNet also have multiple versions. Second, in our experience, we have already used the diverse versions of Meta-Album for various purposes. We anticipate that it will become a widely used benchmark for few-shot-learning applications, with or without meta-learning, and that reviewers with different goals would want different versions. Third, potential users have already expressed interest in having access to the original data, to eventually conduct other preprocessing and data splits, for different purposes.
>
>
> **2. The domains in the existing dataset are relatively close to each other. The authors might want to introduce images from the medical domain such as histopathological images, x-ray, and MRI, to make the tasks even more challenging;**
>
> We very much wish we could. We tried very hard to find suitable datasets in the medical domain, but failed to find datasets meeting all our criteria:
> - Public data that can be re-distributed without restrictions
> - More than 20 classes having at least 40 examples per class
>
> We have 3 datasets in medical domains ready but we cannot release them because of license issues. We would be delighted to get suggestions.

---

### Official Review · Reviewer_BXS8 · 2022-07-22

**Rating:** 7
**Confidence:** 3
**Correctness:** Yes
**Clarity:** The paper is well written and easy to…

**Strengths:**

Strong points:
- The dataset is well-processed and formatted.
- The paper is well written and organized.
- The guidance and the code in github is easy to follow.



**Weaknesses:**

Weak points:
- Meta-Album formats all images into 128$\times$128 pixel, how does it influence the difficulity of the meta learning tasks? Please ensure that it doesn't influence the quality of the image data.
- I'm curious about how does the performance of the meta-trained model vary in different domains? In other word, how to know which domain is more difficult.

**Additional Feedback:**

Please see the weaknesses.

**Documentation:**

The guidance is easy to follow.

**Relation To Prior Work:**

Yes.

**Summary And Contributions:**

This paper introduced an image classification meta-dataset named Meta-Album for cross-domain few-shot learning. It is consisted of the image data collected from 10 different domains, and organized into three versions (miro/mini/extended) . The dataset is well-processed and formatted, and the guidance is easy to follow.

---

> ### Author Response · Authors · 2022-08-06
> **Reviewer's feedback addressed**
>
> We thank the reviewer for their valuable feedback.
>
> ### **Weaknesses:**
>
> **1. Meta-Album formats all images into 128 ×128 pixel, how does it influence the difficulty of the meta learning tasks? Please ensure that it doesn't influence the quality of the image data.**
>
> We ran an evaluation for various resolutions of images, based on the results (highlighted in green in Appendix C.3). According to the preliminary experiments we conducted, the resolution 128 x 128 offers a good compromise between computational burden and loss of accuracy. We chose to carry out the experiments on the insect dataset for which the subject of interest is relatively small and details matter (stripes, legs, antennas).
>
>
> **2. I'm curious about how does the performance of the meta-trained model vary in different domains? In other word, how to know which domain is more difficult.**
>
> This is an excellent point. Indeed, we already started to evaluate dataset difficulty as part of the NeurIPS’22 cross-domain meta-learning challenge, currently on-going, which uses Meta-Album. These results will appear in a paper on the design and baseline results of the challenge, which will appear in PMLR and we’ll refer to it in the Meta-Album paper. A preprint can be found at [https://drive.google.com/file/d/145t-KVmHNIFCweiIjbPwimmAXMvHHf7e/view]. See Figure 3, evaluating Set1. We are currently running the same experiments for all sets and hope to be able to add them in an appendix of the Meta-Album paper.

---

> > ### Author Response · Authors · 2022-08-28
> > **Assessment of datasets' difficulty**
> >
> > We have added a detailed analysis of the difficulty per domain in each set of datasets of Meta-Album. Details in **Appendix D** (in latest version of supplementary material).

---

### Official Review · Reviewer_nJhz · 2022-07-23
**Review of the Meta-Album**

**Rating:** 8
**Confidence:** 3
**Clarity:** This paper is well-written and clear …

**Strengths:**

1.	The dataset is large and well-organized, stemming from multiple domains with various scales and retaining class hierarchy annotations.
2.	It is extensible and growing.
3.	It comprises 3 different versions: micro, mini, and extended, which could well match users’ computational resources.
4.	The authors adequately compare their work and the prior works.


**Weaknesses:**

1.	This work can be improved by comparing more state-of-the-art cross-domain few-shot approaches, such as [1][2][3][4], instead of only classic in-domain few-shot methods.
2.	I would encourage the authors to check the spellings, such as “protocl” in section 2.1.

[1] Li P, Gong S, Wang C, et al. Ranking Distance Calibration for Cross-Domain Few-Shot Learning[C]//Proceedings of the IEEE/CVF Conference on Computer Vision and Pattern Recognition. 2022: 9099-9108.
[2] Li W H, Liu X, Bilen H. Cross-domain Few-shot Learning with Task-specific Adapters[C]//Proceedings of the IEEE/CVF Conference on Computer Vision and Pattern Recognition. 2022: 7161-7170.
[3] Dvornik, N., Schmid, C., and Mairal, J. Selecting relevant features from a multi-domain representation for few-shot classification. In European Conference on Computer Vision . Springer, Cham. 2020: 769-786.
[4] Yue, X et al. "Prototypical cross-domain self-supervised learning for few-shot unsupervised domain adaptation."  Proceedings of the IEEE/CVF Conference on Computer Vision and Pattern Recognition. 2021.


**Additional Feedback:**

Please refer to the Weakness.

**Correctness:**

It would be better if more cross-domain few-shot approaches are conducted on the proposed dataset. Please refer to the Weakness.

**Documentation:**

This work provides sufficient details.

**Ethics:**

This work makes visual inspection of the image quality and ensures there is no offensive material.

**Relation To Prior Work:**

The manuscript clearly discussed its differences and connections with prior works in detail.

**Summary And Contributions:**

This manuscript introduces an image classification dataset for few-shot learning, transfer learning, and other tasks. It contains multiple domains and features various scales, with large numbers of images. Extensive experiments are conducted on the proposed dataset to accomplish few-shot learning under standard 5-way 5-shot setting, cross-domain setting and any-way any-shot setting, proving its practicability.

---

> ### Author Response · Authors · 2022-08-06
> **Reviewer's feedback addressed**
>
> We thank the reviewer for their valuable feedback.
>
> ### **Weaknesses:**
>
> **1. This work can be improved by comparing more state-of-the-art cross-domain few-shot approaches, such as [1][2][3][4], instead of only classic in-domain few-shot methods.**
>
> Thank you for pointing out to us such valuable references. Reviewer TQfC has a similar comment, see also our other reply. We are updating our bibliography to incorporate more state-of-the-art methods. However, (1) the point of our D&B track paper is to make a new meta-dataset available, to enable extensive benchmarks, not to conduct one ourselves necessarily as part of this paper; (2) adding more experiments in the Meta-Album paper is beyond the computational resources we have, because we cannot afford the computers that were used to carry out the original experiments (that could be done with a generous donation of cloud credits by Google).
>
> Nevertheless, we studied the methods you are proposing, to evaluate whether we could accommodate your request. Our findings are that:
>
> [1] uses unlabeled query data, therefore is not compatible with our fully supervised setting.
>
> [2] uses one backbone per domain, then merges them. We estimated it would take around 71.12 hours to run. This would not allow us to run it within the time budget of 45 hours allocated to other methods. We do hope to run subsequent benchmarks with larger time budgets.
>
> [3] also uses one backbone per domain and a selector that chooses between the feature representations of the backbones. We estimated it would take around 62.86 hours to run. Hence, like method [2], comparisons cannot be conducted with the same time budget as other methods.
>
> [4] tackles a different problem (few-shot unsupervised domain adaptation), not compatible with our fully supervised few-shot learning setting.
>
> As our meta-dataset is a growing effort, we ourselves plan to conduct more diverse benchmarks in the future. Moreover, we hope that my open-sourcing Meta-Album and providing community support, others will also be encouraged to benchmark other settings we did not yet consider, such as larger time budgets allowing for multiple-experts, and making use of unlabeled data of self-supervised learning.

---

### Official Review · Reviewer_TQfC · 2022-07-27
**Interesting diverse meta learning benchmark with easy to access versions**

**Rating:** 7
**Confidence:** 3
**Correctness:** Yes, The benchmark dataset is constru…
**Clarity:** The paper is well written and easy to…

**Strengths:**

1. Large number of uniformly formatted datasets from multiple domains
2. Multiple versions with varying sizes of the dataset is provided for better accessibility
3. Baseline methods like ProtoNet, MAML and MatchingNet are evaluated and analysed.
4. Apart from more standard N-way k-shot baseline on any-way any-shot is also provided.

**Weaknesses:**

1. Baseline on more SOTA approaches like iMAML, will be good to have.
2. Domain specific benchmarking is missing, for example how different baseline methods perform on different domains.

**Additional Feedback:**

N/A

**Documentation:**

There is sufficient detail on data collection, availability and maintenance.

**Ethics:**

I believe these are no ethical issues.

**Relation To Prior Work:**

The paper clearly states how it is different from previous datasets in meta-learning.


**Summary And Contributions:**

The authors present Meta-Album, a few-shot image classification benchmark dataset. Meta-Album consists a collection of 40 datasets from multiple domains. All the datasets are pre-processed for accessibility and uniform benchmarking. The authors provide 3 versions (Micro, Mini, Extended) of Meta-Album to facilitate in low resource setting. Further, the Extended version contains more than 1.5M images from 10 domains. Evaluation with benchmark methods like ProtoNet, MAML is given.

---

> ### Author Response · Authors · 2022-08-06
> **Reviewer's feedback addressed**
>
> ### **Weaknesses:**
>
> **1. Baseline on more SOTA approaches like iMAML, will be good to have.**
>
> We thank the reviewer for pointing to this method. We are updating in the Meta-Album paper our review of the state-of-the-art to reference a broader spectrum of SOTA methods and outline their differences. This will be an entry point to Meta-Album users who would want to conduct more in-depth comparative studies and benchmarks (which is indeed one of the uses for Meta-Album). The purpose of our baseline methods is to give a set of “classical” and  “representative” techniques, not to be exhaustive. Since we conducted this study, we no longer have access to the same computers (which were available thanks to a generous donation of Google cloud units), thus a more extensive comparison is beyond what we can offer at this stage. However, we are conducting, in the framework of the NeurIPS’22 cross-domain meta-learning challenge (based on Meta-Album) https://codalab.lisn.upsaclay.fr/competitions/3627 a more extensive comparison, in the exact setting of the challenge, using the computational resources of the challenge. The results will be published separately, as part of the NeurIPS competitions track post-challenge analyses.
>
> **2. Domain specific benchmarking is missing, for example how different baseline methods perform on different domains.**
>
> Maybe we do not understand the remark or the expectations of the reviewer. However, we would like to point to Section 3.1 in which we have specifically conducted a domain specific  benchmark, which we have named: “Within domain few-shot learning”, highlighted in green.

---

### Official Review · Reviewer_2gxR · 2022-07-28
**Meaningful Few-shot Benchmark**

**Rating:** 7
**Confidence:** 4
**Correctness:** Yes.
**Clarity:** Yes.

**Strengths:**

Strength:
1. The benchmark seems to be constructed soundly and the competition is well-organized.
2. Many design decisions in this benchmark point in the right direction, for example, different versions of the challenge for different computation resources, community-driven benchmark, etc.
3. Empirical discoveries in section 3.2 are informative for future research.


**Weaknesses:**

Weaknesses:
1. As vision-language pretraining is developing rapidly, it may be beneficial to explicitly consider a track of cross-domain zero-shot setting or few-shot setting with text information and vision language pretrained models.
2. Uniformly scaling images to the same size may create some difficulty for more fine-grained datasets as they may need larger resolution to provide the necessary cues. However, the design decision is understandable.


**Additional Feedback:**

Overall the reviewer believes the benchmark is very meaningful to the community. Some minor concerns are listed in the weakness section.

**Documentation:**

Yes.

**Ethics:**

No ethics issue.

**Relation To Prior Work:**

Yes.

**Summary And Contributions:**

The paper proposes a large-scale meta-dataset that consists of 30 datasets from 10 domains. Extensive competition and challenges have been organized on the benchmark. Currently it focuses on few-shot classification but in the future it has the potential to be used towards the study of other meaningful problem on learning a general visual perception model.

---

> ### Author Response · Authors · 2022-08-06
> **Reviewer's feedback addressed**
>
> We thank the reviewer for their valuable feedback.
>
> ### **Weaknesses**:
>
> **1. As vision-language pretraining is developing rapidly, it may be beneficial to explicitly consider a track of cross-domain zero-shot setting or few-shot setting with text information and vision language pretrained models.**
>
> This is indeed interesting and valuable. At the moment, the Meta-Album collection does not yet include datasets with extra annotations, which would allow us to address such tasks. We are considering adding more annotations in future releases, depending on resource availability. Or, Meta-Album being extensible and publicly available, we hope that the community (including experts in cross-modal problems) will make such contributions, which we highly encourage.
>
> **2. Uniformly scaling images to the same size may create some difficulty for more fine-grained datasets as they may need larger resolution to provide the necessary cues. However, the design decision is understandable.**
>
> We acknowledge that this choice is limiting to some extent. Our main motivation is that it facilitates rapid research iterations and the organization of competitions with numerous participants and a limited computational budget. In addition, we provide instructions to the community to retrieve versions of the datasets that have the original image resolutions. We conducted preliminary experiments (reported in appendix, see Appendix C) in which we evaluated that the chosen resolution is adequate for the problems at hand (i.e., texture-based patterns of single centered objects). Finally, the chosen resolution (128x128) is a big step forward compared to popular benchmarks such as CIFAR-10 and CIFAR-100 (32x32), so we hope that this will raise the bar for a large portion of the community without introducing computational thresholds.

---

### Meta-Review · Area_Chair_EgRW · 2022-09-07

**Recommendation:** Accept
**Confidence:** 4

**Metareview:**

The reviewers were unanimous in recommending acceptance for this paper. Multiple reviewers commented on the dataset/competition being well organized, and the experimental baselines provided in the paper were generally appreciated. There were some suggestions of expanding the baselines in the paper or further diversifying the domains; I tend to agree that these would be nice, but not necessary for the current paper's acceptance. I recommend acceptance.

---

### Decision · Program_Chairs · 2022-09-16

Accept